# Empirical Localization of Homogeneous Divergences on Discrete Sample Spaces

**Takashi Takenouchi**
Department of Complex and Intelligent Systems
Future University Hakodate
116-2 Kamedanakano, Hakodate, Hokkaido, 040-8655, Japan
ttakashi@fun.ac.jp

**Takafumi Kanamori**
Department of Computer Science and Mathematical Informatics
Nagoya University
Furocho, Chikusaku, Nagoya 464-8601, Japan
kanamori@is.nagoya-u.ac.jp

## Abstract

In this paper, we propose a novel parameter estimator for probabilistic models on discrete space. The proposed estimator is derived from minimization of homogeneous divergence and can be constructed without calculation of the normalization constant, which is frequently infeasible for models in the discrete space. We investigate statistical properties of the proposed estimator such as consistency and asymptotic normality, and reveal a relationship with the information geometry. Some experiments show that the proposed estimator attains comparable performance to the maximum likelihood estimator with drastically lower computational cost.

## 1 Introduction

Parameter estimation of probabilistic models on discrete space is a popular and important issue in the fields of machine learning and pattern recognition. For example, the Boltzmann machine (with hidden variables) [1] [2] [3] is a very popular probabilistic model to represent binary variables, and attracts increasing attention in the context of Deep learning [4]. A training of the Boltzmann machine, *i.e.*, estimation of parameters is usually done by the maximum likelihood estimation (MLE). The MLE for the Boltzmann machine cannot be explicitly solved and the gradient-based optimization is frequently used. A difficulty of the gradient-based optimization is that the calculation of the gradient requires calculation of a normalization constant or a partition function in each step of the optimization and its computational cost is sometimes exponential order. The problem of computational cost is common to the other probabilistic models on discrete spaces and various kinds of approximation methods have been proposed to solve the difficulty. One approach tries to approximate the probabilistic model by a tractable model by the mean-field approximation, which considers a model assuming independence of variables [5]. Another approach such as the contrastive divergence [6] avoids the exponential time calculation by the Markov Chain Monte Carlo (MCMC) sampling.

In the literature of parameters estimation of probabilistic model for continuous variables, [7] employs a score function which is a gradient of log-density with respect to the data vector rather than parameters. This approach makes it possible to estimate parameters without calculating the normalization term by focusing on the shape of the density function. [8] extended the method to discrete variables, which defines information of "neighbor" by contrasting probability with that of a flipped

variable. [9] proposed a generalized local scoring rules on discrete sample spaces and [10] proposed an approximated estimator with the Bregman divergence.

In this paper, we propose a novel parameter estimator for models on discrete space, which does not require calculation of the normalization constant. The proposed estimator is defined by minimization of a risk function derived by an unnormalized model and the homogeneous divergence having a weak coincidence axiom. The derived risk function is convex for various kind of models including higher order Boltzmann machine. We investigate statistical properties of the proposed estimator such as the consistency and reveal a relationship between the proposed estimator and the $\alpha$-divergence [11].

## 2 Settings

Let $X$ be a $d$-dimensional vector of random variables in a discrete space $\mathcal{X}$ (typically $\{+1, -1\}^d$) and a bracket $\langle f \rangle$ be summation of a function $f(\boldsymbol{x})$ on $\mathcal{X}$, i.e., $\langle f \rangle = \sum_{\boldsymbol{x} \in \mathcal{X}} f(\boldsymbol{x})$. Let $\mathcal{M}$ and $\mathcal{P}$ be a space of all non-negative finite measures on $\mathcal{X}$ and a subspace consisting of all probability measures on $\mathcal{X}$, respectively.

$$\mathcal{M} = \{f(\boldsymbol{x}) \,|\, \langle f \rangle < \infty, f(\boldsymbol{x}) \geq 0\}, \ \mathcal{P} = \{f(\boldsymbol{x}) \,|\, \langle f \rangle = 1, f(\boldsymbol{x}) \geq 0\}.$$

In this paper, we focus on parameter estimation of a probabilistic model $\bar{q}_{\boldsymbol{\theta}}(\boldsymbol{x})$ on $\mathcal{X}$, written as

$$\bar{q}_{\boldsymbol{\theta}}(\boldsymbol{x}) = \frac{q_{\boldsymbol{\theta}}(\boldsymbol{x})}{Z_{\boldsymbol{\theta}}} \tag{1}$$

where $\boldsymbol{\theta}$ is an $m$-dimensional vector of parameters, $q_{\boldsymbol{\theta}}(\boldsymbol{x})$ is an unnormalized model in $\mathcal{M}$ and $Z_{\boldsymbol{\theta}} = \langle q_{\boldsymbol{\theta}} \rangle$ is a normalization constant. A computation of the normalization constant $Z_{\boldsymbol{\theta}}$ sometimes requires calculation of exponential order and is sometimes difficult for models on the discrete space. Note that the unnormalized model $q_{\boldsymbol{\theta}}(\boldsymbol{x})$ is not normalized and $\langle q_{\boldsymbol{\theta}} \rangle = \sum_{\boldsymbol{x} \in \mathcal{X}} q_{\boldsymbol{\theta}}(\boldsymbol{x}) = 1$ does not necessarily hold. Let $\psi_{\boldsymbol{\theta}}(\boldsymbol{x})$ be a function on $\mathcal{X}$ and throughout the paper, we assume without loss of generality that the unnormalized model $q_{\boldsymbol{\theta}}(\boldsymbol{x})$ can be written as

$$q_{\boldsymbol{\theta}}(\boldsymbol{x}) = \exp(\psi_{\boldsymbol{\theta}}(\boldsymbol{x})). \tag{2}$$

**Remark 1.** *By setting $\psi_{\boldsymbol{\theta}}(\boldsymbol{x})$ as $\psi_{\boldsymbol{\theta}}(\boldsymbol{x}) - \log Z_{\boldsymbol{\theta}}$, the normalized model* (1) *can be written as* (2).

**Example 1.** *The Bernoulli distribution on $\mathcal{X} = \{+1, -1\}$ is a simplest example of the probabilistic model* (1) *with the function $\psi_{\theta}(x) = \theta x$.*

**Example 2.** *With a function $\psi_{\boldsymbol{\theta},k}(\boldsymbol{x}) = (x_1, \ldots, x_d, x_1 x_2, \ldots, x_{d-1} x_d, x_1 x_2 x_3, \ldots)\boldsymbol{\theta}$, we can define a $k$-th order Boltzmann machine [1, 12].*

**Example 3.** *Let $\boldsymbol{x}_o \in \{+1, -1\}^{d_1}$ and $\boldsymbol{x}_h \in \{+1, -1\}^{d_2}$ be an observed vector and hidden vector, respectively, and $\boldsymbol{x} = (\boldsymbol{x}_o^T, \boldsymbol{x}_h^T) \in \{+1, -1\}^{d_1+d_2}$ where $T$ indicates the transpose, be a concatenated vector. A function $\psi_{h,\boldsymbol{\theta}}(\boldsymbol{x}_o)$ for the Boltzmann machine with hidden variables is written as*

$$\psi_{h,\boldsymbol{\theta}}(\boldsymbol{x}_o) = \log \sum_{\boldsymbol{x}_h} \exp(\psi_{\boldsymbol{\theta},2}(\boldsymbol{x})), \tag{3}$$

*where $\sum_{\boldsymbol{x}_h}$ is the summation with respect to the hidden variable $\boldsymbol{x}_h$.*

Let us assume that a dataset $\mathcal{D} = \{\boldsymbol{x}_i\}_{i=1}^n$ generated by an underlying distribution $p(\boldsymbol{x})$, is given and $\mathcal{Z}$ be a set of all patterns which appear in the dataset $\mathcal{D}$. An empirical distribution $\tilde{p}(\boldsymbol{x})$ associated with the dataset $\mathcal{D}$ is defined as

$$\tilde{p}(\boldsymbol{x}) = \begin{cases} \frac{n_{\boldsymbol{x}}}{n} & \boldsymbol{x} \in \mathcal{Z}, \\ 0 & \text{otherwise,} \end{cases}$$

where $n_{\boldsymbol{x}} = \sum_{i=1}^n \mathrm{I}(\boldsymbol{x}_i = \boldsymbol{x})$ is a number of pattern $\boldsymbol{x}$ appeared in the dataset $\mathcal{D}$.

**Definition 1.** *For the unnormalized model* (2) *and distributions $p(\boldsymbol{x})$ and $\tilde{p}(\boldsymbol{x})$ in $\mathcal{P}$, probability functions $r_{\alpha,\boldsymbol{\theta}}(\boldsymbol{x})$ and $\tilde{r}_{\alpha,\boldsymbol{\theta}}(\boldsymbol{x})$ on $\mathcal{X}$ are defined by*

$$r_{\alpha,\boldsymbol{\theta}}(\boldsymbol{x}) = \frac{p(\boldsymbol{x})^\alpha q_{\boldsymbol{\theta}}(\boldsymbol{x})^{1-\alpha}}{\langle p^\alpha q_{\boldsymbol{\theta}}^{1-\alpha} \rangle}, \ \tilde{r}_{\alpha,\boldsymbol{\theta}}(\boldsymbol{x}) = \frac{\tilde{p}(\boldsymbol{x})^\alpha q_{\boldsymbol{\theta}}(\boldsymbol{x})^{1-\alpha}}{\langle \tilde{p}^\alpha q_{\boldsymbol{\theta}}^{1-\alpha} \rangle}.$$

The distribution $r_{\alpha,\boldsymbol{\theta}}$ ($\tilde{r}_{\alpha,\boldsymbol{\theta}}$) is an e-mixture model of the unnormalized model (2) and $p(\boldsymbol{x})$ ($\tilde{p}(\boldsymbol{x})$) with ratio $\alpha$ [11].

**Remark 2.** *We observe that $r_{0,\boldsymbol{\theta}}(\boldsymbol{x}) = \tilde{r}_{0,\boldsymbol{\theta}}(\boldsymbol{x}) = \bar{q}_{\boldsymbol{\theta}}(\boldsymbol{x})$, $r_{1,\boldsymbol{\theta}}(\boldsymbol{x}) = p(\boldsymbol{x})$, $\tilde{r}_{1,\boldsymbol{\theta}}(\boldsymbol{x}) = \tilde{p}(\boldsymbol{x})$. Also if $p(\boldsymbol{x}) = \bar{q}_{\boldsymbol{\theta}_0}(\boldsymbol{x})$, $r_{\alpha,\boldsymbol{\theta}_0}(\boldsymbol{x}) = \bar{q}_{\boldsymbol{\theta}_0}(\boldsymbol{x})$ holds for an arbitrary $\alpha$.*

To estimate the parameter $\boldsymbol{\theta}$ of probabilistic model $\bar{q}_{\boldsymbol{\theta}}$, the MLE defined by $\hat{\boldsymbol{\theta}}_{mle} = \operatorname{argmax}_{\boldsymbol{\theta}} L(\boldsymbol{\theta})$ is frequently employed, where $L(\boldsymbol{\theta}) = \sum_{i=1}^{n} \log \bar{q}_{\boldsymbol{\theta}}(\boldsymbol{x}_i)$ is the log-likelihood of the parameter $\boldsymbol{\theta}$ with the model $\bar{q}_{\boldsymbol{\theta}}$. Though the MLE is asymptotically consistent and efficient estimator, a main drawback of the MLE is that computational cost for probabilistic models on the discrete space sometimes becomes exponential. Unfortunately the MLE does not have an explicit solution in general, the estimation of the parameter can be done by the gradient based optimization with a gradient $\langle \tilde{p}\psi'_{\boldsymbol{\theta}} \rangle - \langle \bar{q}_{\boldsymbol{\theta}}\psi'_{\boldsymbol{\theta}} \rangle$ of log-likelihood, where $\psi'_{\boldsymbol{\theta}} = \frac{\partial \psi_{\boldsymbol{\theta}}}{\partial \boldsymbol{\theta}}$. While the first term can be easily calculated, the second term includes calculation of the normalization term $Z_{\boldsymbol{\theta}}$, which requires $2^d$ times summation for $\mathcal{X} = \{+1, -1\}^d$ and is not feasible when $d$ is large.

## 3 Homogeneous Divergences for Statistical Inference

Divergences are an extension of the squared distance and are often used in statistical inference. A formal definition of the divergence $D(f, g)$ is a non-negative valued function on $\mathcal{M} \times \mathcal{M}$ or on $\mathcal{P} \times \mathcal{P}$ such that $D(f, f) = 0$ holds for arbitrary $f$. Many popular divergences such as the Kullback-Leilber (KL) divergence defined on $\mathcal{P} \times \mathcal{P}$ enjoy the coincidence axiom, i.e., $D(f, g) = 0$ leads to $f = g$. The parameter in the statistical model $\bar{q}_{\boldsymbol{\theta}}$ is estimated by minimizing the divergence $D(\tilde{p}, \bar{q}_{\boldsymbol{\theta}})$, with respect to $\boldsymbol{\theta}$.

In the statistical inference using unnormalized models, the coincidence axiom of the divergence is not suitable, since the probability and the unnormalized model do not exactly match in general. Our purpose is to estimate the underlying distribution up to a constant factor using unnormalized models. Hence, divergences having the property of the weak coincidence axiom, i.e., $D(f, g) = 0$ if and only if $g = cf$ for some $c > 0$, are good candidate. As a class of divergences with the weak coincidence axiom, we focus on homogeneous divergences that satisfy the equality $D(f, g) = D(f, cg)$ for any $f, g \in \mathcal{M}$ and any $c > 0$.

A representative of homogeneous divergences is the pseudo-spherical (PS) divergence [13], or in other words, $\gamma$-divergence [14], that is defined from the Hölder inequality. Assume that $\gamma$ is a positive constant. For all non-negative functions $f, g$ in $\mathcal{M}$, the Hölder inequality

$$\left\langle f^{\gamma+1} \right\rangle^{\frac{1}{\gamma+1}} \left\langle g^{\gamma+1} \right\rangle^{\frac{\gamma}{\gamma+1}} - \left\langle fg^{\gamma} \right\rangle \geq 0$$

holds. The inequality becomes an equality if and only if $f$ and $g$ are linearly dependent. The PS-divergence $D_{\gamma}(f, g)$ for $f, g \in \mathcal{M}$ is defined by

$$D_{\gamma}(f, g) = \frac{1}{1+\gamma} \log \left\langle f^{\gamma+1} \right\rangle + \frac{\gamma}{1+\gamma} \log \left\langle g^{\gamma+1} \right\rangle - \log \left\langle fg^{\gamma} \right\rangle, \quad \gamma > 0. \tag{4}$$

The PS divergence is homogeneous, and the Hölder inequality ensures the non-negativity and the weak coincidence axiom of the PS-divergence. One can confirm that the scaled PS-divergence, $\gamma^{-1} D_{\gamma}$, converges to the extended KL-divergence defined on $\mathcal{M} \times \mathcal{M}$, as $\gamma \to 0$. The PS-divergence is used to obtain a robust estimator [14].

As shown in (4), the standard PS-divergence from the empirical distribution $\tilde{p}$ to the unnormalized model $q_{\boldsymbol{\theta}}$ requires the computation of $\langle q_{\boldsymbol{\theta}}^{\gamma+1} \rangle$, that may be infeasible in our setup. To circumvent such an expensive computation, we employ a trick and substitute a model $\tilde{p}q_{\boldsymbol{\theta}}$ localized by the empirical distribution for $q_{\boldsymbol{\theta}}$, which makes it possible to replace the total sum in $\langle q_{\boldsymbol{\theta}}^{\gamma+1} \rangle$ with the empirical mean. More precisely, let us consider the PS-divergence from $f = (p^{\alpha} q^{1-\alpha})^{\frac{1}{1+\gamma}}$ to $g = (p^{\alpha'} q^{1-\alpha'})^{\frac{1}{1+\gamma}}$ for the probability distribution $p \in \mathcal{P}$ and the unnormalized model $q \in \mathcal{M}$, where $\alpha, \alpha'$ are two distinct real numbers. Then, the divergence vanishes if and only if $p^{\alpha} q^{1-\alpha} \propto p^{\alpha'} q^{1-\alpha'}$, i.e., $q \propto p$. We define the *localized PS-divergence* $S_{\alpha,\alpha',\gamma}(p, q)$ by

$$S_{\alpha,\alpha',\gamma}(p, q) = D_{\gamma}((p^{\alpha} q^{1-\alpha})^{1/(1+\gamma)}, (p^{\alpha'} q^{1-\alpha'})^{1/(1+\gamma)})$$

$$= \frac{1}{1+\gamma} \log \left\langle p^{\alpha} q^{1-\alpha} \right\rangle + \frac{\gamma}{1+\gamma} \log \langle p^{\alpha'} q^{1-\alpha'} \rangle - \log \left\langle p^{\beta} q^{1-\beta} \right\rangle, \tag{5}$$

where $\beta = (\alpha + \gamma\alpha')/(1 + \gamma)$. Substituting the empirical distribution $\tilde{p}$ into $p$, the total sum over $\mathcal{X}$ is replaced with a variant of the empirical mean such as $\langle \tilde{p}^\alpha q^{1-\alpha} \rangle = \sum_{\boldsymbol{x} \in \mathcal{Z}} \left(\frac{n_{\boldsymbol{x}}}{n}\right)^\alpha q^{1-\alpha}(\boldsymbol{x})$ for a non-zero real number $\alpha$. Since $S_{\alpha,\alpha',\gamma}(p,q) = S_{\alpha',\alpha,1/\gamma}(p,q)$ holds, we can assume $\alpha > \alpha'$ without loss of generality. In summary, the conditions of the real parameters $\alpha, \alpha', \gamma$ are given by

$$\gamma > 0, \ \alpha > \alpha', \ \alpha \neq 0, \ \alpha' \neq 0, \ \alpha + \gamma\alpha' \neq 0,$$

where the last condition denotes $\beta \neq 0$.

Let us consider another aspect of the computational issue about the localized PS-divergence. For the probability distribution $p$ and the unnormalized exponential model $q_{\boldsymbol{\theta}}$, we show that the localized PS-divergence $S_{\alpha,\alpha',\gamma}(p, q_{\boldsymbol{\theta}})$ is convex in $\boldsymbol{\theta}$, when the parameters $\alpha, \alpha'$ and $\gamma$ are properly chosen.

**Theorem 1.** *Let $p \in \mathcal{P}$ be any probability distribution, and let $q_{\boldsymbol{\theta}}$ be the unnormalized exponential model $q_{\boldsymbol{\theta}}(\boldsymbol{x}) = \exp(\boldsymbol{\theta}^T \boldsymbol{\phi}(\boldsymbol{x}))$, where $\boldsymbol{\phi}(\boldsymbol{x})$ is any vector-valued function corresponding to the sufficient statistic in the (normalized) exponential model $\bar{q}_{\boldsymbol{\theta}}$. For a given $\beta$, the localized PS-divergence $S_{\alpha,\alpha',\gamma}(p, q_{\boldsymbol{\theta}})$ is convex in $\boldsymbol{\theta}$ for any $\alpha, \alpha', \gamma$ satisfying $\beta = (\alpha + \gamma\alpha')/(1 + \gamma)$ if and only if $\beta = 1$.*

*Proof.* After some calculation, we have $\frac{\partial^2 \log\langle p^\alpha q_{\boldsymbol{\theta}}^{1-\alpha}\rangle}{\partial\boldsymbol{\theta}\partial\boldsymbol{\theta}^T} = (1-\alpha)^2 V_{r_{\alpha,\boldsymbol{\theta}}}[\boldsymbol{\phi}]$, where $V_{r_{\alpha,\boldsymbol{\theta}}}[\boldsymbol{\phi}]$ is the covariance matrix of $\boldsymbol{\phi}(\boldsymbol{x})$ under the probability $r_{\alpha,\boldsymbol{\theta}}(\boldsymbol{x})$. Thus, the Hessian matrix of $S_{\alpha,\alpha',\gamma}(p, q_{\boldsymbol{\theta}})$ is written as

$$\frac{\partial^2}{\partial\boldsymbol{\theta}\partial\boldsymbol{\theta}^T} S_{\alpha,\alpha',\gamma}(p, q_{\boldsymbol{\theta}}) = \frac{(1-\alpha)^2}{1+\gamma} V_{r_{\alpha,\boldsymbol{\theta}}}[\boldsymbol{\phi}] + \frac{\gamma(1-\alpha')^2}{1+\gamma} V_{r_{\alpha',\boldsymbol{\theta}}}[\boldsymbol{\phi}] - (1-\beta)^2 V_{r_{\beta,\boldsymbol{\theta}}}[\boldsymbol{\phi}].$$

The Hessian matrix is non-negative definite if $\beta = 1$. The converse direction is deferred to the supplementary material. $\square$

Up to a constant factor, the localized PS-divergence with $\beta = 1$ characterized by Theorem 1 is denotes as $S_{\alpha,\alpha'}(p,q)$ that is defined by

$$S_{\alpha,\alpha'}(p,q) = \frac{1}{\alpha-1} \log \langle p^\alpha q^{1-\alpha}\rangle + \frac{1}{1-\alpha'} \log\langle p^{\alpha'} q^{1-\alpha'}\rangle$$

for $\alpha > 1 > \alpha' \neq 0$. The parameter $\alpha'$ can be negative if $p$ is positive on $\mathcal{X}$. Clearly, $S_{\alpha,\alpha'}(p,q)$ satisfies the homogeneity and the weak coincidence axiom as well as $S_{\alpha,\alpha',\gamma}(p,q)$.

## 4 Estimation with the localized pseudo-spherical divergence

Given the empirical distribution $\tilde{p}$ and the unnormalized model $q_{\boldsymbol{\theta}}$, we define a novel estimator with the localized PS-divergence $S_{\alpha,\alpha',\gamma}$ (or $S_{\alpha,\alpha'}$). Though the localized PS-divergence plugged-in the empirical distribution is not well-defined when $\alpha' < 0$, we can formally define the following estimator by restricting the domain $\mathcal{X}$ to the observed set of examples $\mathcal{Z}$, even for negative $\alpha'$:

$$\hat{\boldsymbol{\theta}} = \underset{\boldsymbol{\theta}}{\arg\min} \, S_{\alpha,\alpha',\gamma}(\tilde{p}, q_{\boldsymbol{\theta}}) \quad (6)$$

$$= \underset{\boldsymbol{\theta}}{\arg\min} \, \frac{1}{1+\gamma} \log \sum_{\boldsymbol{x} \in \mathcal{Z}} \left(\frac{n_{\boldsymbol{x}}}{n}\right)^\alpha q_{\boldsymbol{\theta}}(\boldsymbol{x})^{1-\alpha} + \frac{\gamma}{1+\gamma} \log \sum_{\boldsymbol{x} \in \mathcal{Z}} \left(\frac{n_{\boldsymbol{x}}}{n}\right)^{\alpha'} q_{\boldsymbol{\theta}}(\boldsymbol{x})^{1-\alpha'}$$

$$- \log \sum_{\boldsymbol{x} \in \mathcal{Z}} \left(\frac{n_{\boldsymbol{x}}}{n}\right)^\beta q_{\boldsymbol{\theta}}(\boldsymbol{x})^{1-\beta}.$$

**Remark 3.** *The summation in (6) is defined on $\mathcal{Z}$ and then is computable even when $\alpha, \alpha', \beta < 0$. Also the summation includes only $\mathcal{Z}(\leq n)$ terms and its computational cost is $\mathcal{O}(n)$.*

**Proposition 1.** *For the unnormalized model (2), the estimator (6) is Fisher consistent.*

*Proof.* We observe

$$\left.\frac{\partial}{\partial\boldsymbol{\theta}} S_{\alpha,\alpha',\gamma}(\bar{q}_{\boldsymbol{\theta}_0}, q_{\boldsymbol{\theta}})\right|_{\boldsymbol{\theta}=\boldsymbol{\theta}_0} = \left(\beta - \frac{\alpha+\gamma\alpha'}{1+\gamma}\right)\langle\bar{q}_{\boldsymbol{\theta}_0}\psi'_{\boldsymbol{\theta}_0}\rangle = 0$$

implying the Fisher consistency of $\hat{\boldsymbol{\theta}}$. $\square$

**Theorem 2.** *Let $q_{\boldsymbol{\theta}}(\boldsymbol{x})$ be the unnormalized model (2), and $\boldsymbol{\theta}_0$ be the true parameter of underlying distribution $p(\boldsymbol{x}) = \bar{q}_{\boldsymbol{\theta}_0}(\boldsymbol{x})$. Then an asymptotic distribution of the estimator (6) is written as*

$$\sqrt{n}(\hat{\boldsymbol{\theta}} - \boldsymbol{\theta}_0) \sim \mathcal{N}(\boldsymbol{0}, I(\boldsymbol{\theta}_0)^{-1})$$

*where $I(\boldsymbol{\theta}_0) = V_{\bar{q}_{\boldsymbol{\theta}_0}}[\psi'_{\boldsymbol{\theta}_0}]$ is the Fisher information matrix.*

*Proof.* We shall sketch a proof and the detailed proof is given in supplementary material. Let us assume that the empirical distribution is written as

$$\tilde{p}(\boldsymbol{x}) = \bar{q}_{\boldsymbol{\theta}_0}(\boldsymbol{x}) + \epsilon(\boldsymbol{x}).$$

Note that $\langle \epsilon \rangle = 0$ because $\tilde{p}, \bar{q}_{\boldsymbol{\theta}_0} \in \mathcal{P}$. The asymptotic expansion of the equilibrium condition for the estimator (6) around $\boldsymbol{\theta} = \boldsymbol{\theta}_0$ leads to

$$0 = \frac{\partial}{\partial \boldsymbol{\theta}} S_{\alpha, \alpha', \gamma}(\tilde{p}, q_{\boldsymbol{\theta}}) \Big|_{\boldsymbol{\theta} = \hat{\boldsymbol{\theta}}}$$

$$= \frac{\partial}{\partial \boldsymbol{\theta}} S_{\alpha, \alpha', \gamma}(\tilde{p}, q_{\boldsymbol{\theta}}) \Big|_{\boldsymbol{\theta} = \boldsymbol{\theta}_0} + \frac{\partial^2}{\partial \boldsymbol{\theta} \partial \boldsymbol{\theta}^T} S_{\alpha, \alpha', \gamma}(\tilde{p}, q_{\boldsymbol{\theta}}) \Big|_{\boldsymbol{\theta} = \boldsymbol{\theta}_0} (\hat{\boldsymbol{\theta}} - \boldsymbol{\theta}_0) + \mathcal{O}(||\hat{\boldsymbol{\theta}} - \boldsymbol{\theta}_0||^2)$$

By the delta method [15], we have

$$\frac{\partial}{\partial \boldsymbol{\theta}} S_{\alpha, \alpha', \gamma}(\tilde{p}, q_{\boldsymbol{\theta}}) \Big|_{\boldsymbol{\theta} = \boldsymbol{\theta}_0} - \frac{\partial}{\partial \boldsymbol{\theta}} S_{\alpha, \alpha', \gamma}(p, q_{\boldsymbol{\theta}}) \Big|_{\boldsymbol{\theta} = \boldsymbol{\theta}_0} \simeq -\frac{\gamma}{(1 + \gamma)^2} (\alpha - \alpha')^2 \langle \psi'_{\boldsymbol{\theta}_0} \epsilon \rangle$$

and from the central limit theorem, we observe that

$$\sqrt{n} \langle \psi'_{\boldsymbol{\theta}_0} \epsilon \rangle = \sqrt{n} \frac{1}{n} \sum_{i=1}^{n} \left( \psi'_{\boldsymbol{\theta}_0}(\boldsymbol{x}_i) - \langle \bar{q}_{\boldsymbol{\theta}_0} \psi'_{\boldsymbol{\theta}_0} \rangle \right)$$

asymptotically follows the normal distribution with mean $\boldsymbol{0}$, and variance $I(\boldsymbol{\theta}_0) = V_{\bar{q}_{\boldsymbol{\theta}_0}}[\psi'_{\boldsymbol{\theta}_0}]$, which is known as the Fisher information matrix. Also from the law of large numbers, we have

$$\frac{\partial^2}{\partial \boldsymbol{\theta} \partial \boldsymbol{\theta}^T} S_{\alpha, \alpha', \gamma}(\tilde{p}, q_{\boldsymbol{\theta}}) \Big|_{\boldsymbol{\theta} = \boldsymbol{\theta}_0} (\hat{\boldsymbol{\theta}} - \boldsymbol{\theta}_0) \to \frac{\gamma}{(1 + \gamma)^2} (\alpha - \alpha')^2 I(\boldsymbol{\theta}_0),$$

in the limit of $n \to \infty$. Consequently, we observe that (2). $\qquad\square$

**Remark 4.** *The asymptotic distribution of (6) is equal to that of the MLE, and its variance does not depend on $\alpha, \alpha', \gamma$.*

**Remark 5.** *As shown in Remark 1, the normalized model (1) is a special case of the unnormalized model (2) and then Theorem 2 holds for the normalized model.*

## 5 Characterization of localized pseudo-spherical divergence $S_{\alpha, \alpha'}$

Throughout this section, we assume that $\beta = 1$ holds and investigate properties of the localized PS-divergence $S_{\alpha, \alpha'}$. We discuss influence of selection of $\alpha, \alpha'$ and characterization of the localized PS-divergence $S_{\alpha, \alpha'}$ in the following subsections.

### 5.1 Influence of selection of $\alpha, \alpha'$

We investigate influence of selection of $\alpha, \alpha'$ for the localized PS-divergence $S_{\alpha, \alpha'}$ with a view of the estimating equation. The estimator $\hat{\boldsymbol{\theta}}$ derived from $S_{\alpha, \alpha'}$ satisfies

$$\frac{\partial S_{\alpha, \alpha'}(\tilde{p}, q_{\boldsymbol{\theta}})}{\partial \boldsymbol{\theta}} \Big|_{\boldsymbol{\theta} = \hat{\boldsymbol{\theta}}} \propto \left\langle \tilde{r}_{\alpha', \hat{\boldsymbol{\theta}}} \psi'_{\hat{\boldsymbol{\theta}}} \right\rangle - \left\langle \tilde{r}_{\alpha, \hat{\boldsymbol{\theta}}} \psi'_{\hat{\boldsymbol{\theta}}} \right\rangle = 0. \qquad (7)$$

which is a moment matching with respect to two distributions $\tilde{r}_{\alpha, \boldsymbol{\theta}}$ and $\tilde{r}_{\alpha', \boldsymbol{\theta}}$ $(\alpha, \alpha' \neq 0, 1)$. On the other hand, the estimating equation of the MLE is written as

$$\frac{\partial L(\boldsymbol{\theta})}{\partial \boldsymbol{\theta}} \Big|_{\boldsymbol{\theta} = \boldsymbol{\theta}_{mle}} \propto \langle \tilde{p} \psi'_{\boldsymbol{\theta}_{mle}} \rangle - \langle \bar{q}_{\boldsymbol{\theta}_{mle}} \psi_{\boldsymbol{\theta}_{mle}} \rangle = \left\langle \tilde{r}_{1, \boldsymbol{\theta}_{mle}} \psi'_{\boldsymbol{\theta}_{mle}} \right\rangle - \left\langle \tilde{r}_{0, \boldsymbol{\theta}_{mle}} \psi'_{\boldsymbol{\theta}_{mle}} \right\rangle = 0, \quad (8)$$

which is a moment matching with respect to the empirical distribution $\tilde{p} = \tilde{r}_{1, \boldsymbol{\theta}_{mle}}$ and the normalized model $\bar{q}_{\boldsymbol{\theta}} = \tilde{r}_{0, \boldsymbol{\theta}_{mle}}$. While the localized PS-divergence $S_{\alpha, \alpha'}$ is not defined with $(\alpha, \alpha') = (0, 1)$, comparison of (7) with (8) implies that behavior the estimator $\hat{\boldsymbol{\theta}}$ becomes similar to that of the MLE in the limit of $\alpha \to 1$ and $\alpha' \to 0$.

## 5.2 Relationship with the $\alpha$-divergence

The $\alpha$-divergence between two positive measures $f, g \in \mathcal{M}$ is defined as

$$D_\alpha(f, g) = \frac{1}{\alpha(1-\alpha)} \left\langle \alpha f + (1-\alpha)g - f^\alpha g^{1-\alpha} \right\rangle,$$

where $\alpha$ is a real number. Note that $D_\alpha(f, g) \geq 0$ and $0$ if and only if $f = g$, and the $\alpha$-divergence reduces to $\mathrm{KL}(f, g)$ and $\mathrm{KL}(g, f)$ in the limit of $\alpha \to 1$ and $0$, respectively.

**Remark 6.** *An estimator defined by minimizing $\alpha$-divergence $D_\alpha(\tilde{p}, \bar{q}_{\boldsymbol{\theta}})$ between the empirical distribution and normalized model, satisfies*

$$\frac{\partial D_\alpha(\tilde{p}, \bar{q}_{\boldsymbol{\theta}})}{\partial \boldsymbol{\theta}} \propto \left\langle \tilde{p}^\alpha q_{\boldsymbol{\theta}}^{1-\alpha} \left( \psi'_{\boldsymbol{\theta}} - \langle \bar{q}_{\boldsymbol{\theta}} \psi'_{\boldsymbol{\theta}} \rangle \right) \right\rangle = 0$$

*and requires calculation proportional to $|\mathcal{X}|$ which is infeasible. Also the same hold for an estimator defined by minimizing $\alpha$-divergence $D_\alpha(\tilde{p}, q_{\boldsymbol{\theta}})$ between the empirical distribution and unnormalized model, satisfying $\frac{\partial D_\alpha(\tilde{p}, q_{\boldsymbol{\theta}})}{\partial \boldsymbol{\theta}} \propto \left\langle (1-\alpha)q_{\boldsymbol{\theta}} \psi'_{\boldsymbol{\theta}} - \tilde{p}^\alpha q_{\boldsymbol{\theta}}^{1-\alpha} \right\rangle = 0$.*

Here, we assume that $\alpha, \alpha' \neq 0, 1$ and consider a trick to cancel out the term $\langle g \rangle$ by mixing two $\alpha$-divergences as follows.

$$\begin{aligned} D_{\alpha, \alpha'}(f, g) &= D_\alpha(f, g) + \left( \frac{-\alpha'}{\alpha} \right) D_{\alpha'}(f, g) \\ &= \left\langle \left( \frac{1}{1-\alpha} - \frac{\alpha'}{\alpha(1-\alpha')} \right) f - \frac{1}{\alpha(1-\alpha)} f^\alpha g^{1-\alpha} + \frac{1}{\alpha(1-\alpha')} f^{\alpha'} g^{1-\alpha'} \right\rangle. \end{aligned}$$

**Remark 7.** $D_{\alpha, \alpha'}(f, g) \geq 0$ *is divergence when $\alpha\alpha' < 0$ holds, i.e., $D_{\alpha, \alpha'}(f, g) \geq 0$ and $D_{\alpha, \alpha'}(f, g) = 0$ if and only if $f = g$. Without loss of generality, we assume $\alpha > 0 > \alpha'$ for $D_{\alpha, \alpha'}$.*

Firstly, we consider an estimator defined by the minmizer of

$$\min_{\boldsymbol{\theta}} \sum_{\boldsymbol{x} \in \mathcal{Z}} \left\{ \frac{1}{1-\alpha'} \left( \frac{n_{\boldsymbol{x}}}{n} \right)^{\alpha'} q_{\boldsymbol{\theta}}(\boldsymbol{x})^{1-\alpha'} - \frac{1}{1-\alpha} \left( \frac{n_{\boldsymbol{x}}}{n} \right)^{\alpha} q_{\boldsymbol{\theta}}(\boldsymbol{x})^{1-\alpha} \right\}. \tag{9}$$

Note that the summation in (9) includes only $\mathcal{Z}(\leq n)$ terms. We remark the following.

**Remark 8.** *Let $\bar{q}_{\boldsymbol{\theta}_0}(\boldsymbol{x})$ be the underlying distribution and $q_{\boldsymbol{\theta}}(\boldsymbol{x})$ be the unnormalized model (2). Then an estimator defined by minimizing $D_{\alpha, \alpha'}(\bar{q}_{\boldsymbol{\theta}_0}, q_{\boldsymbol{\theta}})$ is not in general Fisher consistent, i.e.,*

$$\frac{\partial D_{\alpha, \alpha'}(\bar{q}_{\boldsymbol{\theta}_0}, q_{\boldsymbol{\theta}})}{\partial \boldsymbol{\theta}} \bigg|_{\boldsymbol{\theta} = \boldsymbol{\theta}_0} \propto \left\langle \bar{q}_{\boldsymbol{\theta}_0}^{\alpha'} q_{\boldsymbol{\theta}_0}^{1-\alpha'} \psi'_{\boldsymbol{\theta}_0} - \bar{q}_{\boldsymbol{\theta}_0}^{\alpha} q_{\boldsymbol{\theta}_0}^{1-\alpha} \psi'_{\boldsymbol{\theta}_0} \right\rangle = \left( \langle q_{\boldsymbol{\theta}_0} \rangle^{-\alpha'} - \langle q_{\boldsymbol{\theta}_0} \rangle^{-\alpha} \right) \langle q_{\boldsymbol{\theta}_0} \psi'_{\boldsymbol{\theta}_0} \rangle \neq 0.$$

This remark shows that an estimator associated with $D_{\alpha, \alpha'}(\tilde{p}, q_{\boldsymbol{\theta}})$ does not have suitable properties such as (asymptotic) unbiasedness and consistency while required computational cost is drastically reduced. Intuitively, this is because the (mixture of) $\alpha$-divergence satisfies the coincidence axiom.

To overcome this drawback, we consider the following minimization problem for estimation of the parameter $\boldsymbol{\theta}$ of model $\bar{q}_{\boldsymbol{\theta}}(\boldsymbol{x})$.

$$(\hat{\boldsymbol{\theta}}, \hat{r}) = \underset{\boldsymbol{\theta}, r}{\operatorname{argmin}} \, D_{\alpha, \alpha'}(\tilde{p}, r q_{\boldsymbol{\theta}})$$

where $r$ is a constant corresponding to an inverse of the normalization term $Z_{\boldsymbol{\theta}} = \langle q_{\boldsymbol{\theta}} \rangle$.

**Proposition 2.** *Let $q_{\boldsymbol{\theta}}(\boldsymbol{x})$ be the unnormalized model (2). For $\alpha > 1$ and $0 > \alpha'$, the minimization of $D_{\alpha, \alpha'}(\tilde{p}, r q_{\boldsymbol{\theta}})$ is equivalent to the minimization of*

$$S_{\alpha, \alpha'}(\tilde{p}, q_{\boldsymbol{\theta}}).$$

*Proof.* For a given $\boldsymbol{\theta}$, we observe that

$$\hat{r}_{\boldsymbol{\theta}} = \underset{r}{\operatorname{argmin}} \, D_{\alpha, \alpha'}(\tilde{p}, r q_{\boldsymbol{\theta}}) = \left( \frac{\langle \tilde{p}^\alpha q_{\boldsymbol{\theta}}^{1-\alpha} \rangle}{\langle \tilde{p}^{\alpha'} q_{\boldsymbol{\theta}}^{1-\alpha'} \rangle} \right)^{\frac{1}{\alpha - \alpha'}}. \tag{10}$$

Note that computation of (10) requires only sample order $\mathcal{O}(n)$ calculation. By plugging (10) into $D_{\alpha,\alpha'}(\tilde{p}, rq_{\boldsymbol{\theta}})$, we observe

$$\hat{\boldsymbol{\theta}} = \underset{\boldsymbol{\theta}}{\mathrm{argmin}} \, D_{\alpha,\alpha'}(\tilde{p}, \hat{r}_{\boldsymbol{\theta}} q_{\boldsymbol{\theta}}) = \underset{\boldsymbol{\theta}}{\mathrm{argmin}} \, S_{\alpha,\alpha'}(\tilde{p}, q_{\boldsymbol{\theta}}). \qquad (11)$$

$\square$

If $\alpha > 1$ and $\alpha' < 0$ hold, the estimator (11) is equivalent to the estimator associated with the localized PS-divergence $S_{\alpha,\alpha'}$, implying that $S_{\alpha,\alpha'}$ is characterized by the mixture of $\alpha$-divergences.

**Remark 9.** *From a viewpoint of the information geometry [11], a metric (information geometrical structure) induced by the $\alpha$-divergence is the Fisher metric induced by the KL-divergence. This implies that the estimation based on the (mixture of) $\alpha$-divergence is Fisher efficient and is an intuitive explanation of the Theorem 2. The localized PS divergence $S_{\alpha,\alpha',\gamma}$ and $S_{\alpha,\alpha'}$ with $\alpha\alpha' > 0$ can be interpreted as an extension of the $\alpha$-divergence, which preserves Fisher efficiency.*

## 6 Experiments

We especially focus on a setting of $\beta = 1$, *i.e.*, convexity of the risk function with the unnormalized model $\exp(\boldsymbol{\theta}^T \boldsymbol{\phi}(\boldsymbol{x}))$ holds (Theorem 1) and examined performance of the proposed estimator.

### 6.1 Fully visible Boltzmann machine

In the first experiment, we compared the proposed estimator with parameter settings $(\alpha, \alpha') = (1.01, 0.01), (1.01, -0.01), (2, -1)$, with the MLE and the ratio matching method [8]. Note that the ratio matching method also does not require calculation of the normalization constant, and the proposed method with $(\alpha, \alpha') = (1.01, \pm 0.01)$ may behave like the MLE as discussed in section 5.1.

All methods were optimized with the *optim* function in R language [16]. The dimension $d$ of input was set to 10 and the synthetic dataset was randomly generated from the second order Boltzmann machine (Example 2) with a parameter $\boldsymbol{\theta}^* \sim \mathcal{N}(\mathbf{0}, I)$. We repeated comparison 50 times and observed averaged performance. Figure 1 (a) shows median of the root mean square errors (RMSEs) between $\boldsymbol{\theta}^*$ and $\hat{\boldsymbol{\theta}}$ of each method over 50 trials, against the number $n$ of examples. We observe that the proposed estimator works well and is superior to the ratio matching method. In this experiment, the MLE outperforms the proposed method contrary to the prediction of Theorem 2. This is because observed patterns were only a small portion of all possible patterns, as shown in Figure 1 (b). Even in such a case, the MLE can take all possible patterns ($2^{10} = 1024$) into account through the normalization term $\log Z_{\boldsymbol{\theta}} \simeq Const + \frac{1}{2}||\boldsymbol{\theta}||^2$ that works like a regularizer. On the other hand, the proposed method genuinely uses only the observed examples, and the asymptotic analysis would not be relevant in this case. Figure 1 (c) shows median of computational time of each method against $n$. The computational time of the MLE does not vary against $n$ because the computational cost is dominated by the calculation of the normalization constant. Both the proposed estimator and the ratio matching method are significantly faster than the MLE, and the ratio matching method is faster than the proposed estimator while the RMSE of the proposed estimator is less than that of the ratio matching.

### 6.2 Boltzmann machine with hidden variables

In this subsection, we applied the proposed estimator for the Boltzmann machine with hidden variables whose associated function is written as (3). The proposed estimator with parameter settings $(\alpha, \alpha') = (1.01, 0.01), (1.01, -0.01), (2, -1)$ was compared with the MLE. The dimension $d_1$ of observed variables was fixed to 10 and $d_2$ of hidden variables was set to 2, and the parameter $\boldsymbol{\theta}^*$ was generated as $\boldsymbol{\theta}^* \sim \mathcal{N}(\mathbf{0}, I)$ including parameters corresponding to hidden variables. Note that the Boltzmann machine with hidden variables is not identifiable and different values of the parameter do not necessarily generate different probability distributions, implying that estimators are influenced by local minimums. Then we measured performance of each estimator by the averaged

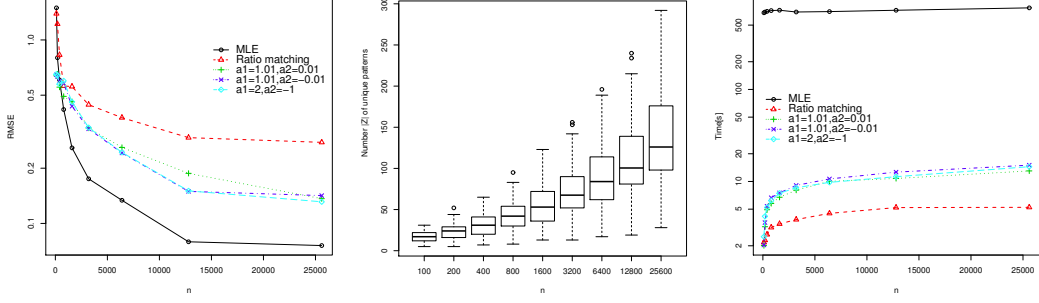

Figure 1: (a) Median of RMSEs of each method against $n$, in log scale. (b) Box-whisker plot of number $|\mathcal{Z}|$ of unique patterns in the dataset $\mathcal{D}$ against $n$. (c) Median of computational time of each method against $n$, in log scale.

log-likelihood $\frac{1}{n} \sum_{i=1}^{n} \log \bar{q}_{\hat{\boldsymbol{\theta}}}(\boldsymbol{x}_i)$ rather than the RMSE. An initial value of the parameter was set by $\mathcal{N}(\boldsymbol{0}, I)$ and commonly used by all methods. We repeated the comparison 50 times and observed the averaged performance. Figure 2 (a) shows median of averaged log-likelihoods of each method over 50 trials, against the number $n$ of example. We observe that the proposed estimator is comparable with the MLE when the number $n$ of examples becomes large. Note that the averaged log-likelihood of MLE once decreases when $n$ is samll, and this is due to overfitting of the model. Figure 2 (b) shows median of averaged log-likelihoods of each method for test dataset consists of 10000 examples, over 50 trials. Figure 2 (c) shows median of computational time of each method against $n$, and we observe that the proposed estimator is significantly faster than the MLE.

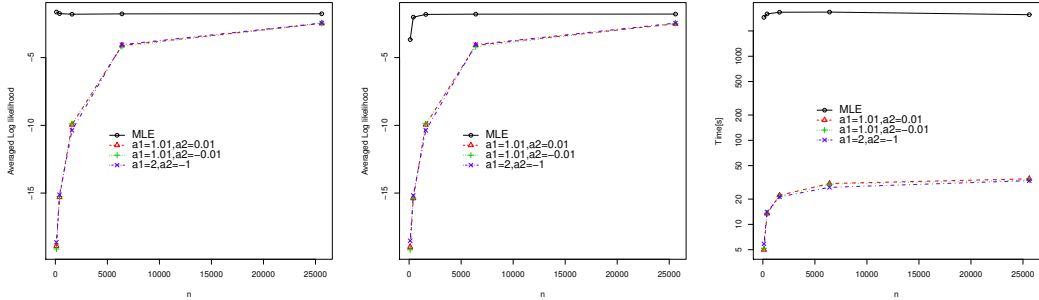

Figure 2: (a) Median of averaged log-likelihoods of each method against $n$. (b) Median of averaged log-likelihoods of each method calculated for test dataset against $n$. (c) Median of computational time of each method against $n$, in log scale.

# 7 Conclusions

We proposed a novel estimator for probabilistic model on discrete space, based on the unnormalized model and the localized PS-divergence which has the homogeneous property. The proposed estimator can be constructed without calculation of the normalization constant and is asymptotically efficient, which is the most important virtue of the proposed estimator. Numerical experiments show that the proposed estimator is comparable to the MLE and required computational cost is drastically reduced.

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
