[Supplementary Material]

# Supplementary Material of "Empirical Localization of Homogeneous Divergences on Discrete Sample Spaces"

## 1  Proof of Theorem 1

The Hessian matrix of $S_{\alpha,\alpha',\gamma}(p, q_{\boldsymbol{\theta}})$ is

$$\frac{\partial^2}{\partial\boldsymbol{\theta}\partial\boldsymbol{\theta}^T} S_{\alpha,\alpha',\gamma}(p, q_{\boldsymbol{\theta}}) = \frac{(1-\alpha)^2}{1+\gamma} V_{r_{\alpha,\boldsymbol{\theta}}}[\boldsymbol{\phi}] + \frac{\gamma(1-\alpha')^2}{1+\gamma} V_{r_{\alpha',\boldsymbol{\theta}}}[\boldsymbol{\phi}] - (1-\beta)^2 V_{r_{\beta,\boldsymbol{\theta}}}[\boldsymbol{\phi}].$$

For a given $\beta \neq 1$, we prove that there exists a model $q_{\boldsymbol{\theta}}$ and parameters $\alpha, \alpha', \gamma$ such that the Hessian is not non-negative definite.

Suppose $\mathcal{X} = \{+1, -1\}^d$. For $\boldsymbol{x} = (x_1, \ldots, x_d) \in \mathcal{X}$, the function $\boldsymbol{\phi}(\boldsymbol{x}) = (\phi_1(\boldsymbol{x}), \ldots, \phi_d(\boldsymbol{x})) \in \mathbb{R}^d$ is defined by $\phi_k(\boldsymbol{x}) = x_k, \ k = 1, \ldots, d$. Then the normalized model is $\bar{q}_{\boldsymbol{\theta}}(\boldsymbol{x}) = \exp\{\sum_{k=1}^d \theta_k x_k - \sum_{k=1}^d \log(e^{\theta_k} + e^{-\theta_k})\}$. Let $p$ be the uniform distribution on $\mathcal{X}$. The covariance matrix of $\boldsymbol{\phi}$ is the diagonal matrix given by

$$V_{r_{\alpha,\boldsymbol{\theta}}}[\boldsymbol{\phi}] = 4 \cdot \mathrm{diag}\left(\frac{1}{(e^{(1-\alpha)\theta_1} + e^{-(1-\alpha)\theta_1})^2}, \cdots, \frac{1}{(e^{(1-\alpha)\theta_d} + e^{-(1-\alpha)\theta_d})^2}\right).$$

Let $\delta$ be $\delta = 1/(1+\gamma)$, then $\delta \in (0, 1)$ holds for $\gamma > 0$. We define

$$f(z; \theta) = \frac{(1-z)^2}{(e^{(1-z)\theta} + e^{-(1-z)\theta})^2}, \quad z, \theta \in \mathbb{R}.$$

Then, the $i$-th diagonal elements of the Hessian matrix is expressed by

$$\Delta = \delta \cdot f(\alpha; \theta_i) + (1-\delta) \cdot f(\alpha'; \theta_i) - f(\delta\alpha + (1-\delta)\alpha'; \theta_i)$$

up to a positive constant. Our task is to find the parameter $\alpha, \alpha', \delta$ such that $\beta = \delta\alpha + (1-\delta)\alpha'$ and $\Delta < 0$ hold. The function $f$ satisfies the following properties.

(a) $f(z; \theta) \geq 0$ and $f(z; \theta) = 0 \Leftrightarrow z = 1$.

(b) $f(1+\varepsilon; \theta) = f(1-\varepsilon; \theta) = f(1+\varepsilon; -\theta)$ for $\varepsilon \geq 0, \theta \in \mathbb{R}$.

(c) $\lim_{z \to \pm\infty} f(z; \theta) = 0$ holds for $\theta \neq 0$.

Let $\theta$ be a fixed non-zero real number. Since $\beta \neq 1$, $f(\beta; \theta) > 0$ holds. Due to the properties (b) and (c), any sufficiently large $\varepsilon > 0$ satisfies $f(1-\varepsilon; \theta) = f(1+\varepsilon; \theta) < f(\beta; \theta)$. Define $\alpha = 1+\varepsilon$ and $\alpha' = 1-\varepsilon$. By choosing $\delta \in (0, 1)$ such that $\beta = \delta\alpha + (1-\delta)\alpha'$, we have $\Delta < 0$.

**Remark 1.** *Even when $\alpha$ and $\alpha'$ are both restricted to positive numbers, we need $\beta = 1$ to ensure the non-negative definiteness of the Hessian matrix. Let us prove this fact. Suppose that $\beta > 1$. Due to (a), (c) in the above and the continuity of $f(z, \theta)$ at $z = 1$, there exists $\alpha$ and $\alpha'$ satisfying $1 < \alpha' < \beta < \alpha$ such that both $f(\alpha'; \theta)$ and $f(\alpha; \theta)$ are less than $f(\beta; \theta)$, where $\theta$ is a non-zero constant. Then, $\Delta < 0$ holds for $\delta \in (0, 1)$ such that $\beta = \delta\alpha + (1-\delta)\alpha'$. We prove the case of $0 < \beta < 1$. For a sufficiently large $\theta$, we have*

$$\frac{f(0; \theta)}{f(\beta; \theta)} = O\left(\frac{e^{-2\beta\theta}}{(1-\beta)^2}\right) \to 0 \ (\theta \to \infty).$$

*Hence, the continuity of $f$ ensures that there exist a sufficiently large $\theta$ and a small positive $\alpha'$ such that $0 < \alpha' < \beta$ and $f(\alpha'; \theta) < f(\beta; \theta)$ hold. The property (c) ensures that there exists a sufficiently large $\alpha > 1$ satisfying $f(\alpha; \theta) < f(\beta; \theta)$. Again, $\Delta < 0$ holds for $\delta$ such that $\beta = \delta\alpha + (1-\delta)\alpha'$.*

## 2  Proof of Theorem 2

Let us assume that the empirical distribution is written as

$$\tilde{p}(\boldsymbol{x}) = \bar{q}_{\boldsymbol{\theta}_0}(\boldsymbol{x}) + \epsilon(\boldsymbol{x}).$$

Note that $\langle \epsilon \rangle = 0$ because $\tilde{p}, \bar{q}_{\boldsymbol{\theta}_0} \in \mathcal{P}$. By expanding an equilibrium condition of the estimator (6) around $\boldsymbol{\theta} = \boldsymbol{\theta}_0$ and $\epsilon(\boldsymbol{x}) = 0$, we obtain

$$0 = \frac{\partial}{\partial \boldsymbol{\theta}} S_{\alpha,\alpha',\gamma}(\tilde{p}, q_{\boldsymbol{\theta}}) \bigg|_{\boldsymbol{\theta}=\hat{\boldsymbol{\theta}}}$$

$$= \frac{\partial}{\partial \boldsymbol{\theta}} S_{\alpha,\alpha',\gamma}(\tilde{p}, q_{\boldsymbol{\theta}}) \bigg|_{\boldsymbol{\theta}=\boldsymbol{\theta}_0} + \frac{\partial^2}{\partial \boldsymbol{\theta} \partial \boldsymbol{\theta}^T} S_{\alpha,\alpha',\gamma}(\tilde{p}, q_{\boldsymbol{\theta}}) \bigg|_{\boldsymbol{\theta}=\boldsymbol{\theta}_0} (\hat{\boldsymbol{\theta}} - \boldsymbol{\theta}_0) + \mathcal{O}(||\hat{\boldsymbol{\theta}} - \boldsymbol{\theta}_0||^2)$$

$$= \left\{ \frac{1-\alpha}{1+\gamma} \left\langle \tilde{r}_{\alpha,\boldsymbol{\theta}_0} \psi'_{\boldsymbol{\theta}_0} \right\rangle + \frac{\gamma(1-\alpha')}{1+\gamma} \left\langle \tilde{r}_{\alpha',\boldsymbol{\theta}_0} \psi'_{\boldsymbol{\theta}_0} \right\rangle - (1-\beta) \left\langle \tilde{r}_{\beta,\boldsymbol{\theta}_0} \psi'_{\boldsymbol{\theta}_0} \right\rangle \right\}$$

$$+ \left\{ \frac{(1-\alpha)^2}{1+\gamma} V_{\tilde{r}_{\alpha,\boldsymbol{\theta}_0}}[\psi'_{\boldsymbol{\theta}_0}] + \frac{\gamma(1-\alpha')^2}{1+\gamma} V_{\tilde{r}_{\alpha',\boldsymbol{\theta}_0}}[\psi'_{\boldsymbol{\theta}_0}] - (1-\beta)^2 V_{\tilde{r}_{\beta,\boldsymbol{\theta}_0}}[\psi'_{\boldsymbol{\theta}_0}] \right.$$

$$\left. + \frac{1-\alpha}{1+\gamma} \left\langle \tilde{r}_{\alpha,\boldsymbol{\theta}_0} \psi''_{\boldsymbol{\theta}_0} \right\rangle + \frac{\gamma(1-\alpha')}{1+\gamma} \left\langle \tilde{r}_{\alpha',\boldsymbol{\theta}_0} \psi''_{\boldsymbol{\theta}_0} \right\rangle - (1-\beta) \left\langle \tilde{r}_{\beta,\boldsymbol{\theta}_0} \psi''_{\boldsymbol{\theta}_0} \right\rangle \right\} (\hat{\boldsymbol{\theta}} - \boldsymbol{\theta}_0) + \mathcal{O}(||\hat{\boldsymbol{\theta}} - \boldsymbol{\theta}_0||^2).$$

By the delta method [1], we observe that

$$(\langle \tilde{r}_{\alpha,\boldsymbol{\theta}_0} \psi'_{\boldsymbol{\theta}_0} \rangle - \langle r_{\alpha,\boldsymbol{\theta}_0} \psi'_{\boldsymbol{\theta}_0} \rangle) \simeq \alpha \frac{\left\langle \bar{q}_{\boldsymbol{\theta}_0}^{\alpha-1} q_{\boldsymbol{\theta}_0}^{1-\alpha} \psi'_{\boldsymbol{\theta}_0} \epsilon \right\rangle \left\langle \bar{q}_{\boldsymbol{\theta}_0}^{\alpha} q_{\boldsymbol{\theta}_0}^{1-\alpha} \right\rangle - \left\langle \bar{q}_{\boldsymbol{\theta}_0}^{\alpha} q_{\boldsymbol{\theta}_0}^{1-\alpha} \psi'_{\boldsymbol{\theta}_0} \right\rangle \left\langle \bar{q}_{\boldsymbol{\theta}_0}^{\alpha-1} q_{\boldsymbol{\theta}_0}^{1-\alpha} \epsilon \right\rangle}{\left\langle \bar{q}_{\boldsymbol{\theta}_0}^{\alpha} q_{\boldsymbol{\theta}_0}^{1-\alpha} \right\rangle^2}$$

$$= \alpha \left( \langle \psi'_{\boldsymbol{\theta}_0} \epsilon \rangle - \left\langle \bar{q}_{\boldsymbol{\theta}_0} \psi'_{\boldsymbol{\theta}_0} \right\rangle \langle \epsilon \rangle \right)$$

$$= \alpha \langle \psi'_{\boldsymbol{\theta}_0} \epsilon \rangle.$$

The last equality comes from $\langle \epsilon \rangle = \sum_{\boldsymbol{x} \in \mathcal{X}} \epsilon(\boldsymbol{x}) = 0$. Then we have

$$\frac{\partial}{\partial \boldsymbol{\theta}} S_{\alpha,\alpha',\gamma}(\tilde{p}, q_{\boldsymbol{\theta}}) \bigg|_{\boldsymbol{\theta}=\boldsymbol{\theta}_0} - \frac{\partial}{\partial \boldsymbol{\theta}} S_{\alpha,\alpha',\gamma}(p, q_{\boldsymbol{\theta}}) \bigg|_{\boldsymbol{\theta}=\boldsymbol{\theta}_0}$$

$$= \frac{1-\alpha}{1+\gamma} \left( \langle \tilde{r}_{\alpha,\boldsymbol{\theta}_0} \psi'_{\boldsymbol{\theta}_0} \rangle - \langle r_{\alpha,\boldsymbol{\theta}_0} \psi'_{\boldsymbol{\theta}_0} \rangle \right) + \frac{\gamma(1-\alpha')}{1+\gamma} \left( \langle \tilde{r}_{\alpha',\boldsymbol{\theta}_0} \psi'_{\boldsymbol{\theta}_0} \rangle - \langle r_{\alpha',\boldsymbol{\theta}_0} \psi'_{\boldsymbol{\theta}_0} \rangle \right) - (1-\beta) \left( \langle \tilde{r}_{\beta,\boldsymbol{\theta}_0} \psi'_{\boldsymbol{\theta}_0} \rangle - \langle r_{\beta,\boldsymbol{\theta}_0} \psi'_{\boldsymbol{\theta}_0} \rangle \right)$$

$$= \left\{ \frac{1-\alpha}{1+\gamma} \langle \tilde{r}_{\alpha,\boldsymbol{\theta}_0} \psi'_{\boldsymbol{\theta}_0} \rangle + \frac{\gamma(1-\alpha')}{1+\gamma} \langle \tilde{r}_{\alpha',\boldsymbol{\theta}_0} \psi'_{\boldsymbol{\theta}_0} \rangle - (1-\beta) \langle \tilde{r}_{\beta,\boldsymbol{\theta}_0} \psi'_{\boldsymbol{\theta}_0} \rangle \right\} - \left\{ 1 - \frac{\alpha + \gamma\alpha'}{1+\gamma} - (1-\beta) \right\} \langle \bar{q}_{\boldsymbol{\theta}_0} \psi'_{\boldsymbol{\theta}_0} \rangle$$

$$\simeq \left( \alpha \frac{1-\alpha}{1+\gamma} + \alpha' \frac{\gamma(1-\alpha')}{1+\gamma} - (1-\beta)\beta \right) \langle \psi'_{\boldsymbol{\theta}_0} \epsilon \rangle$$

$$= -\frac{\gamma}{(1+\gamma)^2}(\alpha - \alpha')^2 \langle \psi'_{\boldsymbol{\theta}_0} \epsilon \rangle.$$

From the central limit theorem,

$$\sqrt{n} \langle \psi'_{\boldsymbol{\theta}_0} \epsilon \rangle = \sqrt{n} \frac{1}{n} \sum_{i=1}^{n} \left( \psi'_{\boldsymbol{\theta}_0}(\boldsymbol{x}_i) - \langle \bar{q}_{\boldsymbol{\theta}_0} \psi'_{\boldsymbol{\theta}_0} \rangle \right)$$

asymptotically follows the normal distribution with mean $\mathbf{0}$, and variance $V_{\bar{q}_{\boldsymbol{\theta}_0}}[\psi'_{\boldsymbol{\theta}_0}]$, which is known as the Fisher information matrix. Also from the law of large number, we have

$$\frac{(1-\alpha)^2}{1+\gamma} V_{\tilde{r}_\alpha}[\psi'_{\boldsymbol{\theta}_0}] + \frac{\gamma(1-\alpha')^2}{1+\gamma} V_{\tilde{r}_{\alpha'}}[\psi'_{\boldsymbol{\theta}_0}] - (1-\beta)^2 V_{\tilde{r}_\beta}[\psi'_{\boldsymbol{\theta}_0}] \to \frac{\gamma}{(1+\gamma)^2}(\alpha - \alpha')^2 V_{\bar{q}_{\boldsymbol{\theta}_0}}[\psi'_{\boldsymbol{\theta}_0}],$$

$$\frac{1-\alpha}{1+\gamma} \langle \tilde{r}_{\alpha,\boldsymbol{\theta}_0} \psi''_{\boldsymbol{\theta}_0} \rangle + \frac{\gamma(1-\alpha')}{1+\gamma} \langle \tilde{r}_{\alpha',\boldsymbol{\theta}_0} \psi''_{\boldsymbol{\theta}_0} \rangle - (1-\beta) \langle \tilde{r}_{\beta,\boldsymbol{\theta}_0} \psi''_{\boldsymbol{\theta}_0} \rangle \to \left( 1 - \frac{\alpha + \gamma\alpha'}{1+\gamma} - (1-\beta) \right) \langle \bar{q}_{\boldsymbol{\theta}_0} \psi''_{\boldsymbol{\theta}_0} \rangle = 0$$

in the limit of $n \to \infty$. Consequently, we observe that

$$\sqrt{n}(\hat{\boldsymbol{\theta}} - \boldsymbol{\theta}_0) \sim \mathcal{N}(\mathbf{0}, V_{\bar{q}_{\boldsymbol{\theta}_0}}[\psi'_{\boldsymbol{\theta}_0}]^{-1}).$$

# References

[1] A. W. Van der Vaart. *Asymptotic Statistics*. Cambridge University Press, 1998.