[Reviews · NeurIPS 2015]

Submitted by Assigned_Reviewer_1

This paper proposes a new method to derive an estimator for discrete value parameters in probabilistic models, which does not require the calculation of the normalizing constant. This avoids a computational bottle neck, which is often of exponential order and makes many problems infeasible. This new estimator minimizes a risk function defined as the localized pseudo-spherical (PS) divergence between the empirical distribution of the dataset and the unnormalized probabilistic model. This risk function can be made to be convex with properly chosen parameters. The paper shows that its proposed estimator is asymptotically efficient and additionally gives an intuitive explanation of this by showing the localized PS divergence can be interpreted as a mixture of alpha-divergences. The method is applied to fully visible and hidden variable Boltzmann machines and the estimator is compared with the MLE.

The quality of this paper is very good. It is well written, organized, and technically strong. I think there is a typo in Remark 9 on line 337, which should be just alpha > 0 in the parentheses so that sgn(alpha) doesn't matter.

Additionally it would be nice if the simulations compared (both of RMSE and computation time) the proposed estimator to estimators from other methods [5], [6], and [10] instead of just the MLE.

There are several issues that should be dealt with.

In line 352, the experiment on fully observed Boltzmann machine is done with data dimension d=10. This is not very practical. Note that the claim of this paper is the efficiency, so it should tested on a larger dimensional example, e.g. d greater than 100.

Furthermore, depending on the modeling of Boltzmann machine in line 82 and line 88, it is not fully convincing that this method is ready to be applied in deep neural network with large dimension of hidden layer.

Furthermore, the author's result only applies to exponentially families, and may not necessarily hold in a mixture family like the Boltzmann machine with hidden layers.

The critical assumption for the model to work is that the empirical distribution is restricted on a finite subregion $\cZ$, while the model is defined on domain $\set{0,1}^{n} \supset \cZ$.

The authors should make this clear in the statement of their theory.

It is noted that in the last line of page 3, the author introduces a trick to compute the weighted normalization factor as

\sum_{x}\ind{x\in \cZ}\frac{n_{x}}{n}q(x). It should be mentioned that in general

Boltzmann machine is a mixture model that does not satisfy the constraint that there exists a domain Z \subset \set{0,1}^{n}, such that \frac{n_{x}}{n} = 0, x\not\in Z.

There are some other confusing typos:

1. In line 225, r_{\alpha, \theta_{0}}(x) = \bar{q}_{\theta_{0}} (x) should be r_{0}, \theta_{0}}(x) = \bar{q}_{\theta_{0}} (x) as confusing typo. Same typo is in supplementary material.

2. In line 225, "By expanding an equilibrium condition" is not clearly explained in either main text and supplementary material. Could it just be the Taylor expansion?

This should be clarified.

3. Supplementary material line 92, a typo, \langle \phi \epsilon\rangle should be \langle \psi'_{\theta_{0}} \epsilon\rangle.

Summary: This paper present an interesting approach for feasible parameter estimation of discrete probabilistic models. The material is presented in a clear manner, and the only real critique is a comparison to other mentioned estimators for higher dimensional examples would strengthen the paper.

Submitted by Assigned_Reviewer_2

There has recently been a fair amount of work on techniques for estimating exponential family models by minimizing divergence functions, _without_ having to also calculate the normalizing constant or partition function of the model, as would be required for maximum likelihood estimation.

The present MS. extends this to exponential families on discrete sample spaces, particularly Boltzmann machines, with a new-to-me divergence function.

The divergence is built to be well-defined even for unnormalized exponential functions and invariant under scaling; also, the particular divergence to be minimized in estimation is "localized" using the empirical distribution, an interesting technical trick.

The divergence is convex in the parameters of the exponential family, can be calculated in O(N) time, and is claimed to be both consistent and asymptotically efficient.

I say "claimed" in the last sentence because I think the proof of Proposition 1 is too hasty.

What is needed is to show that the sequence of the minimizers of S(\tilde{p}, q_{\theta}) converges on \theta_0.

(I omit the subscripts on S for brevity.)

What is shown is that S(q_{\theta_0}, q_{\theta}) has zero derivative when \theta=\theta_0.

It is true that S is convex in \theta, so this tells us that S(q_{\theta_0}, q_{\theta}) is indeed minimized when \theta=\theta_0.

It's hard for a minimizing estimator to converge otherwise, but this doesn't directly tell us about the convergence of the sequence of estimates.

After all, it is also pretty straightforward to show that KL(q_{\theta_0}, q_{\theta}) has zero derivative when \theta_0=\theta, but showing the convergence of maximum likelihood takes a bit more work than that!

(Cf. van der Vaart, ch. 5.)

Also, in the proof of Theorem 2 in the supplementary material, I am unable to follow the equations which come right after the phrase "By the delta method" (lines 70ff of p. 2) --- they appear to presume that the ratio of two sums is the sum of two ratios.

I am quite prepared to believe I am missing something here but then at the least the exposition needs to be improved.

I also note that the numerical experiment shown in Figure 1 shows the MLE and the new method displaying very different scalings of RMSE with n, contrary to theorem 2; the MS. explains this away due to the support of the empirical distribution being much smaller than that of the true distribution.

I find this explanation obscure --- it doesn't seem to follow from the proof of the theorem --- and the discrepancy disquieting, but not necessarily fatal to the theorem.

To sum up, this is an interesting estimation method, which numerically seems to do well at trading statistical power for speed in large sample spaces.

I think the statistical propositions need either patches to their proofs or re-statements, but I suspect these would be fairly straightforward fixes.
Summary: An interesting extension of recent work on estimating exponential families without needing to calculate normalizing factors / partition functions.

Submitted by Assigned_Reviewer_3

This paper proposes a divergence measure that makes it possible to estimate parameters of unnormalized models (without the need for computing the normalization constant). This is particularly effective for parameter inference of probabilistic models such that the computational complexity of computing the normalization constant is of exponential order, such as the Boltzmann machines. It is also noteworthy in that it has several desirable properties: For unnormalized exponential models, the localized PS divergence with \beta=1 becomes convex in the parameters; The minimum localized PS-divergence estimator has Fisher consistency; When \alpha and \alpha' are set at (\alpha,\alpha') \approx (1,0), the proposed estimator behaves like the ML estimator.

This paper is very interesting, which I truly enjoyed reading. I have some comments and questions, which I hope to be addressed in the resubmitted version.

I take it that the proposed parameter estimation framework can be applied to other probabilistic models than the Boltzmann machines. What else can it be useful for?

I find the notion of the weak coincidence axiom particularly interesting. Is the PS-divergence the only class of divergence measures that hold this property?

The experimental result presented in 6.1 showed that the MLE outperformed the proposed method. The authors state that this is because the number of observed samples was not sufficiently large (and did not comply with the asymptotic analysis). How large should n be to assure that the asymptotic analysis become almost true? I am curious to see whether the performance of the proposed method really approaches that of the MLE when n is larger than 25000.

Quality: Very good Clarity: Excellent Originality: Excellent Significance: Very good
Summary: This paper proposes a divergence measure that makes it possible to estimate parameters of unnormalized models (without the need for computing the normalization constant). This is particularly effective for parameter inference of probabilistic models such that the computational complexity of computing the normalization constant is of exponential order, such as the Boltzmann machines. This paper is very interesting, which I truly enjoyed reading. I have some comments and questions, which I hope to be addressed in the resubmitted version.

Submitted by Assigned_Reviewer_4

The paper proposes a new parameter estimator for probabilistic models on discrete space, which is built on pseudo-spherical divergence, and estimates the unnormalized model.

Estimating the parameters for probabilistic models on discrete space is an important problem and handling the normalization constant is a key difficulty. The proposed estimator doesn't need to compute the normalization constant, and furthermore is convex under mild condition (can choose to design the estimator to satisfy it), which leads to computational advantage.

However, it seems to me the design of the estimator significantly relies on existing work, especially on pseudo-spherical divergence. The paper first focuses on the unnormalized model, which leads to pseudo-spherical divergence. To get rid of the computation of the normalization constant in pseudo-spherical divergence, the estimator is localized to the empirical distribution. This is an interesting idea, but whether it leads to a better estimator is not well illustrated. Consistency/asymptotic normality are established, while finite sample bound is lacking, which can be a great contribution.

Summary: The paper proposes a new parameter estimator for probabilistic models on discrete space. The paper focuses on estimating the unnormalized model, which doesn't require the calculation of the normalization constant.

Author Feedback
Author rebuttal: We thank all the reviewers for detailed comments on our manuscript. Comments by reviewers are summarized to (a) model assumption, (b) simulation results and (c) theorems. The followings are responses to each point.

(a) Model assumptions(rev-1,4)

* models with large dimension of hidden layer (rev-1,4):
Our method is available not only to unnormalized (curved) exponential families but also to unnormalized models expressed by (2), as long as the computation of (2) is tractable. It is true that Boltzmann machines with large dimensional hidden variables suffer from the computational difficulty in general even for unnormalized models, because of the combinatorial computation in (3). However, the restricted Boltzmann machine (RBM), that is an important ingredient of deep neural networks, has a simple form of (3) and the computation complexity of (3) is of the order of the number of hidden variables. Hence, we believe that our method is useful for the pre-training of the RBM; see also the response w.r.t. simulations.

* Z should be a subregion of X (rev-1):
When the domain of the model is very large such as X={-1,1}^d with a large d, usually, the domain of the empirical distribution, Z, is properly included in X. If the domain Z almost covers X, it means that the sample size is extremely large, and a naive application of most statistical methods will not work. In such a situation, we need some heuristics. In the final version, we will give a supplementary explanation about the situation on which our method works.

(b) Simulations(rev-1,2,4,5)

* simulation results in our paper(rev-4,5):
As pointed out by some reviewers, simulation results indicate that the statistical accuracy of the proposed method is not necessarily the same as that of the MLE. In numerical experiments, the difference between our method and the MLE depends on the true parameter of the model. In preliminary experiments, we already confirmed that the proposed method was almost as efficient as the MLE when the true distribution was rather close to the uniform distribution. In such a case, the empirical distribution with a moderate sample size well approximated the target distribution included in a small-scale model. On the contrary, when the target distribution is far from the uniform distribution, the empirical distribution does not necessarily approximate the true one. Even in such a case, the MLE performs well as long as it is computationally feasible. A possible reason is that the normalization term of the probability model behaves like a regularization term. Indeed, for the fully visible Boltzmann machine, the Taylor expansion of the normalization term Z_{theta} up to the second order yields the squared norm of the parameter, i.e., |theta|^2. Hence, the overfitting to training data will be prevented. We have a preliminary result in which the statistical accuracy of our method is improved by employing the regularization technique without additional computational cost. In the final version or longer version of the paper, we will further investigate the localized PS-div with regularization.

* simulation for larger dimensional examples (rev-1), comparison to other methods (rev-1,2):
As we explained in the response w.r.t. model assumptions, our method is applicable to large scale RBMs. After the submission of our NIPS paper, we conducted an experiment for the MNIST dataset (60,000 examples, 28*28=784 dimension) using the RBM with 100 hidden variables, and we obtained some positive results. In the final and/or longer version of the paper, we are planing to include these results in addition to the comparison to other learning methods.

(c) Theorems in the paper (rev-3,5)

* asymptotic consistency and efficiency (rev-5):
Since we skipped the definition of the Fisher consistency, there might be some confusion about the consistency. As pointed out by rev-5, the asymptotic consistency is another important concept to guarantee the statistical accuracy of estimators. Since our theoretical analysis falls into the standard statistical asymptotic theory of M-estimators, showing the so-called "regularity condition" for asymptotic consistency and asymptotic normality is not very hard. In the revised MS, we will show a regularity condition as a supplementary material. In the proof of theorem 2, the calculation is long but straightforward. In the revised MS, more detailed exposition will be presented.

* finite sample bound (rev-3):
As the rev-3 pointed out, the finite sample bound is very important to guarantee the practical statistical performance of learning methods. We admit that there is a gap between the practical situation and theory. The finite sample bound suggested by rev-3 is an important issue, which will be discussed in the revised paper.